# The emergence of cost effective battery storage

Stephen Comello [1] & Stefan Reichelstein[1,2]

Energy storage will be key to overcoming the intermittency and variability of renewable energy sources. Here, we propose a metric for the cost of energy storage and for identifying optimally sized storage systems. The levelized cost of energy storage is the minimum price per kWh that a potential investor requires in order to break even over the entire lifetime of the storage facility. We forecast the dynamics of this cost metric in the context of lithium-ion batteries and demonstrate its usefulness in identifying an optimally sized battery charged by an incumbent solar PV system. Applying the model to residential solar customers in Germany, we find that behind-the-meter storage is economically viable because of the large difference between retail rates and current feed-in tariffs. In contrast, investment incentives for battery systems in California derive principally from a state-level subsidy program.

[1] Stanford Graduate School of Business, 655 Knight Way, Stanford, CA 94305, USA. [2] University of Mannheim, Schloss, Ostfluegel, Mannheim 68131, Germany. Correspondence and requests for materials should be addressed to S.C. (email: scomello@stanford.edu) or to S.R. (email: reichelstein@uni-mannheim.de)

A s the share of renewable energy in the overall energy mix increases, issues of intermittency and dispatchability of the electricity supply are emerging as central in the quest for a grid that is both stable and decarbonized[1–3]. Cost effective energy storage is arguably the main hurdle to overcoming the generation variability of renewables. Though energy storage can be achieved in a variety of ways, battery storage has the advantage that it can be deployed in a modular and distributed fashion[4]. This feature partly explains the recent growth in behind-the-meter storage applications, for instance, when rooftop solar is combined with battery storage[5–11]. Our analysis builds on recent studies that have sought to assess the economic viability of battery storage systems in conjunction with renewable power generation.

The Levelized Cost of Energy Storage (LCOES) metric examined in this paper captures the unit cost of storing energy, subject to the system not charging, or discharging, power beyond its rated capacity at any point in time. This power constraint effectively determines the average duration of the storage system, that is, the average amount of energy that can be stored per kilowatt of power capacity. For any given storage system, the significance of the LCOES metric is that it yields a minimum price that investors would require on average per kWh of electricity stored and subsequently dispatched in order to break even on their investments. In the 2019 market environment for lithium-ion batteries, we estimate an LCOES of around twelve U.S. cents per kWh for a 4-hour duration system, with this cost dropping to ten cents for a 6-hour duration system.

Our analysis demonstrates the use of the LCOES measure in identifying the optimal size of a battery that is charged by an incumbent solar PV system. In the context of residential behind-the-meter storage, the economic benefit of storage capacity is that it yields a price premium, given as the difference between the retail electricity price and the overage tariff that is obtained for surplus energy generated by the solar PV system but not self-consumed. In contrast, the household does not derive additional revenues from reduced demand charges that frequently apply to corporate customers[7,12]. To effectively complement an intermittent solar PV system with storage, an optimally sized battery system will be such that the price premium is equal to the LCOES evaluated at the duration corresponding to the last (marginal) power component. The actual duration of this marginal power component is determined jointly by the solar PV generation and the load consumption profile of the representative household.

We first apply this optimization framework in the context of German households where feed-in tariffs for solar PV power have recently been reduced, yielding a time-invariant price premium around 16 € cents per kWh. We find that this price premium is sufficient to incentivize the installation of a battery, as the levelized cost of storage for the optimally sized system is around 8.5 € cents per kWh, owing to a duration of ~7 h. A significant share of all existing behind-the-meter storage installations in the U.S. are in California[13]. This may seem surprising since California has reaffirmed its commitment to net metering. Taken by itself, net metering would effectively eliminate any price premium, yet the state effectively also created a price premium by imposing a non-bypassable surcharge on electricity consumed from the grid for customers with rooftop solar installations[14,15]. We find that this surcharge in combination with the state's rebate program, titled Self Generation Incentive Program (SGIP), and the federal Investment Tax Credit (ITC) is sufficient to incentivize substantial investments in behind-the meter battery storage. At the same time, the structure of the rebates under SGIP will result in battery systems with a relatively short duration but large power rating.

## Results

**The cost of energy storage.** The primary economic motive for electricity storage is that power is more valuable at times when it is dispatched compared to the hours when the storage device is charged[8,12,16–18]. These benefits will accrue over the entire lifetime of the storage system and must be weighed against the cost of acquiring a system capable of performing the storage service for a given number of charging/discharging events per year over the useful life of the system. A battery will be sized in the two dimensions of power and energy capacity. The size of the power component, measured in kW, governs the maximum rated electricity charge/discharge rate. The energy component determines the total capacity of electricity that can be stored. It is measured in kWh. Moreover, the ratio of energy capacity to rated power determines the duration for which the storage facility can provide the rated power. This is also the length of time needed to charge the battery given its power rating.

To capture the unit cost associated with energy storage, we introduce the Levelized Cost of Energy Storage (LCOES) which, like the commonly known Levelized Cost of Energy, is measured in monetary units (say U.S. $) per kWh. Similar to the LCOE that indicates the average revenue an investor would need in order to break-even over the life cycle of a power generating facility[19], the LCOES measure captures the break-even value for charging and discharging electricity on a per kWh basis.

Earlier studies on the cost of storage have usually fixed the duration of the storage system at some exogenous value, for instance, 4 h[18,20]. In contrast, we first decompose the overall unit cost into the levelized cost of energy components, LCOEC (in $ per kWh), and the levelized cost of power components, LCOPC (in $ per kW). As shown in the Methods section, these levelized costs are obtained by dividing the system price of the power and energy components, respectively, by the total discounted number of charge/discharge occurrences that the battery performs the storage service in the course of its useful life. In particular, the number of charge/discharge events per year is multiplied by a factor that reflects the useful life of the battery, the cost of capital (discount rate), round-trip efficiency losses and, finally, performance degradation over time. It is presumed that these cycling occurrences are not limited to only full capacity charging and discharging, rather, they would include partial charge/discharge events where the capacity of the battery is not fully charged or discharged.

For a storage system with a power rating of $k_p$ kW and a storage capacity of $k_e$ kWh, the corresponding average duration is defined as $D \equiv \frac{k_e}{k_p}$ hours. The duration, $D$, indicates the number of hours that the system charges and discharges $k_p$ kW of power per full cycle. On a lifetime basis, the cost of storing one kWh of electricity, and dispatching it at later hours of the same cycle is LCOES $(D) = $ LCOEC $+ $ LCOPC $\cdot \frac{1}{D}$. Since $D$ is stated in hours, LCOES($\cdot$) is expressed in $ per kWh. The following claim identifies the LCOES metric as the break-even price per kWh for electricity storage services.

Claim: LCOES $\left(\frac{k_e}{k_p}\right) = $ LCOEC $+ $ LCOPC $\cdot \frac{k_p}{k_e}$ is the break-even price for storing and dispatching $k_e$ kWh of energy in $N$ charging/discharging events per year, subject to the maximum power charge (discharge) not exceeding $k_p$ kW at any point in time.

The preceding claim is formally validated in the Methods section below. To calibrate the LCOES metric in the context of lithium-ion, batteries, the following calculations are based on current U.S. market prices of $171 per kWh and $970 per kW for energy and power components, respectively, in the context of small residential applications (see Supplementary Figures 1, 2 and Supplementary Tables 1–3). In the context of lithium-ion batteries, we expand the cost model in order to allow for certain

costs related to installation to be entirely independent of the size of the battery, e.g., permitting, inspecting, and commissioning. In the location-specific application of our model, these fixed costs are estimated to be $400 in the U.S. and $300 (€260) in Germany (see Supplementary Notes 1–6).

Assuming $N = 365$ charging/discharging events, a 10-year useful life of the energy storage component, a 5% cost of capital, a 5% round-trip efficiency loss, and a battery storage capacity degradation rate of 1% annually, the corresponding levelized cost figures are LCOEC = $0.067 per kWh and LCOPC = $0.206 per kW for 2019. The solid curve in Fig. 1 shows the corresponding LCOES for alternative duration values. In the presence of installation related fixed costs, LCOES then yields the break-even price for covering the systems costs that do vary proportionally with either $k_e$ or $k_p$; see Methods section for further details.

Consistent with the recent widespread installation of li-ion based batteries, the LCOES of such systems has dropped dramatically in recent years[21–23]. The dotted curve shows the corresponding nominal LCOES figures back in 2013. Projecting into the future, the consensus forecast that emerges from various literature sources are annual percentage declines of 5.6% and 8.1% for the acquisition cost of the power and storage components, respectively, over the horizon 2018–2023. Assuming that this rate of improvement can indeed be maintained on average during those years, the dashed curve in Fig. 1 provides a forecast of where the LCOES of li-ion battery systems is expected to be in 2023 (see Supplementary Notes 2 and 4).

Our LCOES metric is a variant of existing storage cost measures[18,20,24–27]. For energy generation, the familiar LCOE measure is frequently conceptualized as total (discounted) cash flows spent divided by total (discounted) energy delivered[27,28]. Existing studies on the levelized cost of storage follow the same total-cost-divided-by-total-energy approach[20,26,29,30]. While our LCOES measure is also calibrated as a break-even measure, our metric departs from two individual levelized cost measure (power and energy) and then aggregates these two measures depending on the average duration of the system. This disaggregation will prove useful in characterizing optimally sized storage systems.

In particular, the following section shows that the conditions for an optimally sized battery can be expressed succinctly in terms of the optimized duration. By optimizing the duration of the battery storage system, we obtain cost figures that are consistent with the recent widespread and increasing deployment of such storage systems. Earlier studies that arrived at substantially higher cost of storage have frequently fixed the duration at 2 or 4 h[20,26]. It should also be noted that our LCOES concept only captures the cost per kWh of warehousing electricity on a daily basis, subject to the system's power rating constraint. In contrast, some earlier studies also include the cost of generating the energy that is being dispatched[26,29].

**Optimal battery size supplementing a solar PV system.** We consider a representative household that has already installed a solar PV system and now faces the question whether behind-the meter storage adds additional value. To illustrate the basic economic tradeoff, we first consider a household for which both the consumption load and the solar generation profile are fairly constant across the seasons of the year. The solid curve in Fig. 2 depicts the average load profile and the bell-shaped dotted curve depicts the average solar PV generation curve.

Absent any battery storage, the household will self-consume the energy represented by the area marked I in Fig. 2, and buy the energy outside the time interval $[t^-, t^+]$ at the going retail rate, denoted by $p$. The surplus energy from the solar system, i.e., the regions marked II and III, can possibly be sold back to the energy service provider at some overage tariff, OT which, in Germany, is given by the prevailing feed-in tariff. If a battery system is added, the energy corresponding to the region marked as II in Fig. 2 would be discharged during times when household demand exceeds generation by the rooftop solar facility. Accordingly, region IV in Fig. 2 is equal to region II minus the round-trip efficiency losses. The following derivation also maintains the implicit assumption that the demand represented by combined areas represented IV and V is sufficiently large to absorb the stored energy corresponding to II. Region V is residual demand that would not be met by the battery and must be met through

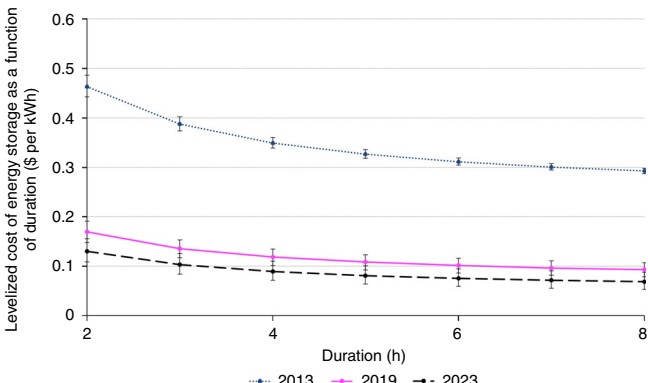

**Fig. 1** Simulated trajectory for lithium-ion LCOES ($ per kWh) as a function of duration (hours) for the years 2013, 2019, and 2023. For energy storage systems based on stationary lithium-ion batteries, the 2019 estimate for the levelized cost of the power component, LCOPC, is $0.206 per kW, while the levelized cost of the energy component, LCOEC, is $0.067 per kWh. The curve corresponding to the year 2019 plots the corresponding LCOES values for alternative levels of the storage system's duration. The LCOES curve corresponding to the year 2013 indicates the decline in lithium-ion based battery storage costs over the past five years. The 2023 curve projects anticipated future cost reductions. Source data are provided as a Source Data file

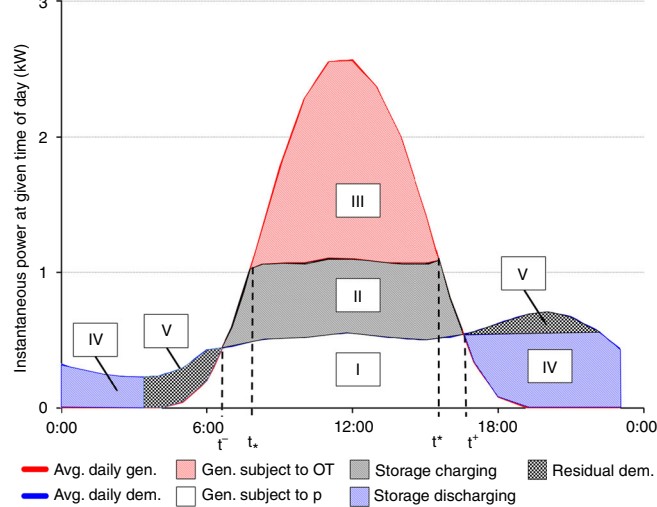

**Fig. 2** Pattern of daily charging and discharging of a battery supplementing a PV system. Region I represents self consumption from solar generation; region II is surplus energy that can be stored and subsequently discharged as region IV (minus efficiency losses); and region III is surplus energy sold to the grid. Region V is residual demand that would not be met by the battery and must be met through purchases from the grid at the going retail rate $p$

purchases from the grid at the going retail rate $p$. The general case —which underlies our calculations and empirical findings—is presented in the Methods section.

A battery supplementing an existing solar rooftop system will add value only if the difference between the retail rate $p$ and the overage tariff, OT, is sufficiently large to cover the levelized cost of the optimally sized battery. With a higher power rating, $k_p$, the amount of energy that can be stored will grow at a diminishing rate (Fig. 2). Referring to the power generation curve as $G(\cdot)$ and the household load curve as $L(\cdot)$, the area (energy) corresponding to region II in Fig. 2 becomes:

$$E^+(k_p) = \int_0^{24} \Big[ \min\{L(t) + k_p, G(t)\} - \min\{L(t), G(t)\} \Big] dt. \tag{1}$$

Our characterization of an optimally sized battery follows the standard microeconomic approach of an (household) investor that seeks to maximize the discounted value of future expected cash flows. As shown in Methods, the overall present value of all cash flows associated with the battery storage system (excluding any fixed costs that do not vary with the size of the battery) is proportional to the profit margin, PM, per cycle, given by:

$$PM(k_p) = [pp - LCOEC] \cdot E^+(k_p) - LCOPC \cdot k_p. \tag{2}$$

Here, pp refers to the price premium, which is the time-averaged difference between $p$ and OT, adjusted for round-trip efficiency losses and the temporal degradation of the energy discharged by the battery. Equation (2) shows that in order for a storage system to add value, the price premium, pp, must exceed the levelized cost of the energy component, LCOEC, yet this is not sufficient because of the need to cover the levelized cost of the power component, LCOPC. The marginal return to systems with a higher power rating is diminishing and therefore a storage system with positive net-present value can be found if, and only if, the daily profit margin is positive for small values of $k_p$. The value of $E^+(k_p)$ is approximately equal to $k_p \cdot (t^+ - t^-)$ for small values of $k_p$ (by L'Hospital's rule). Therefore some storage system will be valuable for the representative household whenever $[pp - LCOEC] \cdot (t^+ - t^-) - LCOPC > 0$.

To identify an optimally sized battery system in this setting, we refer to $\mathcal{I}_+(k_p)$ as the duration of the marginal power component, for a given power rating, $k_p$, of the battery. Formally, $\mathcal{I}^+(k_p)$ is defined as the length of the time interval(s): $I^+(k_p) \equiv \{t \in [0,24] | L(t) + k_p < G(t)\}$. For the battery storage system illustrated in Fig. 2 the duration of the marginal power component is $(t^* - t_*)$ hours. In general, the duration of the marginal power component is the derivative of the function $E^+(\cdot)$ with respect to $k_p$. The first-order optimality condition corresponding to (2) therefore is

$$pp = LCOES(\mathcal{I}^+(k_p^*)), \tag{3}$$

with the optimal storage capacity determined by $E^+(k_p^*)$ according to Eq. (1). Thus the LCOES concept introduced above can be used to identify the size of the value-maximizing battery system by equating the price premium and the LCOES evaluated at the duration of the marginal power component. To illustrate this criterion in the context of Fig. 2, suppose the investor is considering to add an additional kW of power capacity. That would enable the addition of at most another $(t^* - t_*)$ kWh of storage capacity because the solar installation shown in Fig. 2 would make at most that many additional kWh of surplus energy available for charging. The incremental revenue of this addition would therefore be $pp \cdot (t^* - t_*)$ kWh, while the incremental

levelized cost would be LCOPC + LCOEC $\cdot (t^* - t_*) \cdot h$. Optimality requires that the incremental cost be equal to the incremental revenue, or after dividing by $(t^* - t_*) \cdot h$, the optimality condition becomes pp = LCOES $((t^* - t_*) \cdot h)$.

It should be noted that the average duration of the optimal battery storage system exceeds the duration of the marginal power component, because $\frac{\hat{k}_e(k_p^*)}{k_p^*} > \mathcal{I}^+(k_p^*)$. This inequality reflects that the optimal duration, $\frac{\hat{k}_e(k_p^*)}{k_p^*}$, is the average of the durations of the inframarginal power components. In Fig. 2, these durations range from the lowest $(t^* - t_*) \cdot h$ to the maximum value $(t^+ - t^-) \cdot h$.

In Methods we present an extended version of the model that allows for both the solar generation and household electricity demand to vary across the different months of the year. Central to this extension is the variable $E_s(k_p)$. For a given month (season), $s$, the energy quantity $E_s(k_p)$ represents the maximum that can both be charged during the hours of the middle of the day with surplus solar power and discharged in the hours when the household's load exceeds the available solar rooftop energy. The optimally sized battery will generally be oversized relative to the needs of a particular month and undersized in others. As a consequence, the battery will not be fully charged in certain months and therefore will not go through full charging/discharging events for parts of the year. Similarly, it will depend on the month whether the household is grid-positive in the sense that, on a daily basis, the household sells energy to the grid, yet does not buy power from the grid. In contrast, the battery storage system shown in Fig. 2 leaves the household a prosumer—self-producer and grid-consumer of electricity—on account of regions III and V.

Our analysis is complementary to other studies that have explored residential solar PV coupled with storage. Some studies have sought to calculate the levelized cost of storage with solar PV, though with energy and power components the size of which is exogenously fixed rather than determined through an explicit optimization [20,31]. Some studies have determined optimal power and energy dimensions for storage systems of various technologies, though only in the context of a 1-year simulation[10]. Other studies rely on sensitivity analysis to find optimal solar PV plus storage new-build combined systems without explicit consideration of the dimension of the power component or the corresponding life-cycle cost metric[5,16]. Finally, some earlier studies have also focused on sizing a storage system when revenues are obtained through price arbitrage[27], though in contrast to our approach, these studies do not yield estimates for the optimized levelized cost of energy storage.

**Location-specific applications.** Incentives for distributed energy generation in Germany have long been provided by feed-in tariffs. For recent solar installations these tariffs have recently been reduced to ≈12 € cents per kWh[32]. The current retail rates of near 30 € cents therefore create a substantial price premium. In contrast to other jurisdictions around the world, Germany provides only modest direct incentives for battery storage installations[33].

Table 1 shows the results of our model for a representative household in Munich. The calculations are based on a model with 12 representative days, one for each month of the year (see Supplementary Table 4 and Supplementary Note 7). For a $k_s = 6 \, kW_p$ residential solar installation, we obtain an optimal battery size of $k_p^* = 0.73 \, kW$ and $k_e^* = 5.1 \, kWh$, yielding an average duration of about 7 h. The corresponding levelized cost of storage is LCOES = 8.5 € cents per kWh and, separately, the fixed cost is given as €260.

To calibrate our findings, we note that the ratio of $k_s$ to $k_e^*$ aligns with residential battery system sizes observed in

| Table 1 Monthly simulation results for Munich, Germany | | | | | | | | | | | | |
|---|---|---|---|---|---|---|---|---|---|---|---|---|
| | **Jan.** | **Feb.** | **Mar.** | **Apr.** | **May** | **Jun.** | **Jul.** | **Aug.** | **Sep.** | **Oct.** | **Nov.** | **Dec.** |
| $E_s(k_p)$ [kWh] | 3.35 | 5.02 | 6.37 | 5.72 | 5.10 | 4.15 | 4.22 | 4.85 | 5.93 | 5.51 | 3.82 | 1.69 |
| $A_s$ [kWh] | 10.73 | 8.92 | 7.10 | 5.80 | 5.11 | 4.15 | 4.22 | 4.85 | 6.03 | 7.28 | 8.99 | 11.07 |
| Charge to capacity? | No | No | Yes | Yes | Yes | No | No | No | Yes | Yes | No | No |
| Grid-positive? | No | No | No | No | No | Yes | Yes | Yes | No | No | No | No |

Monthly simulation results for Munich, Germany. $A_s$ denotes the aggregate load that is not met by solar generation (regions IV and V in Fig. 2) for that particular month. $E_s(k_p)$ denotes the maximum energy in month $s$ that can be charged during the middle of the day and discharged when the household's load exceeds the available solar energy (formal expression provided in Methods). The optimally sized battery will generally be oversized relative to the needs of a particular month and undersized in others. As a consequence, the battery will not be fully charged (i.e., not charged to capacity) in certain months and therefore will not go through full charging/discharging cycles for parts of the year.

Germany[34]. While our calculations identify the battery capacity yielding the highest net-present value from a pure arbitrage perspective, the household could essentially double the size of the battery and still break-even on the investment. Specifically, the net-present value would still be non-negative if $k_p = 1.6$ kW and $k_e = 8$ kWh. That energy capacity is, in fact, close to the observed average capacity installation in Germany for 2017[34]. Depending on the specific application of the battery system and the frequency of short duration high-power loads, it may be advantageous to opt for a higher power capacity and additional grid purchases.

The optimally sized battery will be fully charged on representative days during five months of the year. Thus $k_e^* = 5.1 \leq E_s(k_p)$. In the remaining 7 months there is either insufficient supply of solar power (November–February) or there is insufficient demand in the after sunset hours (June–August). The household will be grid-positive during average days in June, July, and August in the sense that the energy stored in the battery is sufficient to meet the household's electricity load. Formally, $\min\{k_e^*, E_s(k_p)\} = A_s$ during those months.

Under current rules, solar PV systems in Germany exhaust their feed-in tariff support after 20 years. At that point in time, the household would probably not receive more than the average wholesale rate (around 3 € cents per kWh) for any surplus energy sold back to the energy service provider. For such older solar PV systems, the financial return from adding a battery system would at that point increase further because the price premium would then effectively be at least 27 € cents per kWh. Under this scenario, it would be economical to install a larger power rating (over 4 kW) and coupled larger energy capacity (between 8 €kWh and 11 kWh).

In contrast to Germany, California may at first glance not appear ideally suited for behind-the-meter storage installations. The state's continued commitment to the policy of net metering effectively allows both residential and commercial customers to sell any temporal surplus electricity back to their energy service providers at the same rate at which the customer acquires electricity from the grid. From the perspective of the customer, this policy effectively enables free energy storage. In the context of the above model, the price premium for storage systems in California would therefore be zero. It should be noted, though, that in confirming the state's commitment to net metering, California utilities were nonetheless allowed to impose a non-bypassable charge for all electricity consumed by customers with residential solar PV installations. The average non-bypassable charge rate is 2.7¢ per kWh. This surcharge thus becomes an effective price premium because any energy fed back to the grid from the solar facility will only be credited at the basic electricity retail rate.

For new storage installations in California, investors are eligible for both federal and state-level support programs. At the federal level, battery installations in the U.S. qualify for an Investment Tax Credit, ITC, provided the battery can be classified as solar equipment[35]. Specifically, this requires that the energy storage capability of the battery does not exceed the total energy generated by the solar PV system. As detailed further in Methods, the battery would otherwise only be eligible for a share of the maximum ITC, which amounts to 30% of the acquisition cost.

In addition to the federal support for battery storage systems installed in conjunction with solar PV systems, California has adopted the so-called Self-Generation Incentive Program (SGIP) for behind-the-meter storage systems (see Supplementary Note 8). This program ultimately explains why the majority of residential battery storage systems installed to date in the U.S. are in fact located in California. In its current form, SGIP offers a rebate on the expenditure for energy storage components. Thus it is stated in $ per kWh and effectively reduces the acquisition cost of energy storage components, that is, the parameter $\nu_e$.

The specific amount to be rebated depends on the duration of the storage system. Normalizing $k_p$ at 1 kW, the investor is entitled to a rebate of $400 for the first two kWh of energy storage, an additional rebate of $250 for the next two kWh, and a final rebate of $100 for the next two kWh, up to a duration of 6 h. Additional energy storage components corresponding to the initial 1 kW power rating do not receive any subsidy. For a power component of $k_p$ kW, the rebate amounts to $400 for each kWh of energy storage provided the duration of the system does not exceed 2 h[36]. For systems with longer durations, the rebate per kWh steps down as indicated above, such that no additional support is given to systems with a duration exceeding 6 h. These rebates are available in addition to the federal 30% investment tax credit (ITC) for batteries that qualify as solar equipment.

Table 2 shows the results of our model applied to Los Angeles (see Supplementary Table 5 and Supplementary Note 9). Given a 4.85 kW$_p$ residential solar installation, we obtain an optimal battery size of $k_p^* = 2.45$ kW and $k_e^* = 9.8$ kWh. The optimal duration is $D^* = 4$ h with a corresponding LCOES of 0.6¢ per kWh and, separately, the fixed cost is given as $400.

In contrast to our findings for Germany, the optimally sized battery in California would be charged to capacity effectively every month other than December and January (there is only a small amount of slack of 0.02 kWh in July). At the same time, the representative household will be grid-positive only in the month of June, as in every other month the household will have to continue purchasing power after acquisition of the optimally sized battery. Since the effective price premium is 2.7¢ per kWh, such a system creates value for the investing party under current conditions. Absent any federal or state-level incentives, the LCOES would be 12.8¢ per kWh. The ITC covers 3.8¢ ($\approx 0.3 \cdot$ 12.8¢), while SGIP contributes the remaining 8.4¢.

The fact that the optimal duration in California is exactly 4 h is not a coincidence. It rather reflects that under SGIP the incremental rebate for systems with a duration exceeding 4 h drops from $200 to $100 per kWh. In conjunction with the federal ITC, the California rebate is sufficiently large so as to incentivize a duration between 4 and 6 h, even in the absence of

**Table 2 Monthly simulation results for Los Angeles, California**

|  | Jan. | Feb. | Mar. | Apr. | May | Jun. | Jul. | Aug. | Sep. | Oct. | Nov. | Dec. |
|---|---|---|---|---|---|---|---|---|---|---|---|---|
| $E_s(k_p)$ [kWh] | 7.55 | 10.72 | 12.57 | 10.92 | 10.1 | 9.78 | 11.41 | 12.18 | 11.45 | 10.37 | 9.28 | 7.5 |
| $A_s$ [kWh] | 15.71 | 14.39 | 12.57 | 10.92 | 10.1 | 9.78 | 11.41 | 12.18 | 13.79 | 14.21 | 13.94 | 15.5 |
| Charge to capacity? | No | Yes | Yes | Yes | Yes | No | Yes | Yes | Yes | Yes | Yes | No |
| Grid-positive? | No | No | No | No | No | Yes | No | No | No | No | No | No |

Monthly simulation results for Los Angeles, California. $A_s$ denotes the aggregate load that is not met by solar generation (Areas IV and V in Fig. 2) for that particular month. $E_s(k_p)$ denotes the maximum energy in month s that can be charged during the middle of the day and discharged when the household's load exceeds the available solar energy (formal expression provided in Methods). The optimally sized battery will generally be oversized relative to the needs of a particular month and undersized in others. As a consequence, the battery will not be fully charged (i.e., not charged to capacity) in certain months and therefore will not go through full charging/discharging cycles for parts of the year.

any price premium. The only reason a California household would not install an arbitrarily large storage system is that the full federal tax credit would no longer be available once the energy storage capability exceeds 12.2 kWh, which is the annual average daily production from the solar installation.

We note that an implication of the model is that SGIP could result in economic storage systems (i.e., $PM(k_p) \geq 0$) with an oversized power rating, relative to what is needed to store the peak production surplus from solar generation. The largest system that would be economical to install has the dimensions of $k_p = 2.9$ kW and $k_e = 11.6$ kWh. In order to maintain a relatively short duration and thereby qualify for a correspondingly high rebate under SGIP, the household will acquire a power component that is 20% larger than optimal. To witness, the duration of the optimally sized battery in California is about half of that in Germany. This conclusion needs to be qualified by the observation that our analysis has employed average hourly load data that do not reflect short-term spikes in power consumption. If such spikes occur with sufficient frequency (e.g., electric vehicle charging, simultaneous clothes drying, and air conditioning operation), the household could be left with a tradeoff between greater autonomy resulting from a more powerful battery and additional electricity purchases from the grid.

## Discussion
In order to achieve the renewable energy goals articulated by many jurisdictions around the world, energy storage will need to provide a sizable buffer against the variability and intermittency that is inherent to power generation from wind and solar energy. A frequently articulated concern about energy storage is the common perception of high cost. This paper argues that the cost of storage is driven in large part by the duration of the storage system. Duration, which refers to the average amount of energy that can be (dis)charged for each kW of power capacity, will be chosen optimally depending on the underlying generation profile and the price premium for stored energy. The economies of scale inherent in systems with longer durations apply to any energy storage system. In comparison to battery storage, we would expect these scale economies to be more pronounced for hydrogen storage. There the levelized cost of the electrolyzer that converts electricity and water into hydrogen and oxygen is arguably larger than the levelized cost of the power component of a battery system[37]. In contrast the LCOEC for hydrogen storage is likely to be smaller than that of li-ion cells if the hydrogen is stored in tanks or underground caverns[37].

For lithium-ion batteries, we find that, depending on the duration, an effective upper bound on the current unit cost of storage would be about 27¢ per kWh under current U.S. market conditions. Such a high cost would be obtained for a system with a duration of 1 h, that is, 1 kWh of energy that can be charged, or discharged, in 1 h ($k_p = 1$). In that case, the levelized cost of storage in the current U.S. market would amount to LCOES =

LCOPC + LCOEC = 20.6 + 6.7¢ per kWh. At the same time, our analysis shows that a household in the German setting that seeks to optimize the size of its behind-the-meter storage facility, so as to complement a rooftop solar installation, would find it optimal to adopt a 7 h duration system. Not only will that would reduce the LCOES to 8.5 € cents per kWh, such a larger duration system would experience relatively few full cycling events, prolonging operational life[38,39]. Furthermore, within the Germany case, if the recent trend in the system prices for lithium-ion batteries continues, we project this levelized cost of storage figure to decrease to about 6.5 € cents per kWh within another four years.

For an incumbent solar installation, the levelized cost of energy storage concept is useful in order to identify an optimally sized behind-the-meter battery system. The key observation for storage in conjunction with a solar PV system is that for each additional kW of power capacity, the additional number of hours that the system can absorb energy is diminishing. This leads to the simple optimality condition that the available price premium will be equal to the LCOES evaluated at the duration corresponding to the last power component. This model framework yields new predictions for the storage systems that residential solar customers in both Germany and California would want to adopt. In both jurisdictions, investments in battery storage are now emerging as financially advantageous provided the storage system is sized and configured optimally. Yet, the economic fundamentals differ substantially due to the different regulatory regimes in place.

The incentives to invest in battery storage in Germany are principally driven by the sizable difference between retail rates and feed-in tariffs. The resulting price premium for energy that is self generated and stored of about 16 € cents per kWh generates a tangible profit margin in comparison to the optimized LCOES value of about 8.5 € cents per kWh. The prospect of further reductions in battery acquisition costs in the future as well as the policy trend towards lower feed-in tariffs for residential solar PV power is likely to strengthen the incentives for installing behind-the meter storage devices in the future.

Our explanation for the recent growth in residential storage capacity in California is based primarily on the combination of the state's rebate program in conjunction with the federal investment tax credit that is available for storage devices that qualify as solar equipment. In contrast to Germany, the price premium is modest under California's net-metering policy. As the acquisition cost of storage modules continues to decline, the state of California is likely to adjust the current rebate program. Our analysis suggests that a more cost effective approach for the rebate program would lessen the incentive for storage systems with a relatively short duration, such as the 4 h that emerge as most profitable for the household in our calculations.

There are several promising avenues for extending the analysis in this paper. First, while the analysis in this paper has viewed the adoption of the solar PV system as exogenous and given, many commercial and residential investors will in the future view

distributed generation and storage as a simultaneous investment decision. Depending on the overage tariff that is available for energy that is fed to the grid, the availability of cost effective behind-the-meter storage will provide incentives for a larger PV system, which, in turn, is likely to increase the size of the optimally sized battery system[11,12,40]. Secondly, our framework is also readily extended to commercial rather than residential electricity customers. An additional financial motive to invest in battery storage for commercial customers is the avoidance of demand charges and peak pricing charges under time-of-use pricing. In the U.S., these features are becoming increasingly important in the tariff structures for many commercial customers. Such tariff structures provide additional incentives for behind-the-meter storage installations.

## Methods

**LCOES as the break-even value.** We first demonstrate that $\text{LCOES}\left(\frac{k_e}{k_p}\right) = \text{LCOEC} + \text{LCOPC} \cdot \frac{k_p}{k_e}$ is the break-even value per kWh for investing in an energy storage system. At first, we suppose that all system cost scale linearly with the size of the battery storage system. Specifically, suppose a household or an external service provider invests in a system that is capable of storing and dispatching $k_e$ kWh of energy per cycle, subject to the maximum power charge (or discharge) not exceeding $k_p$ kW at any point in time. Suppose further that there are $N$ cycles per year and the unit revenue for storage is $\$p^o$ per kWh. We follow the standard approach in economics that posits that the investing party (household) will seek to optimize the discounted value of future cash flows (NPV) resulting from investment in a storage system:

$$\text{NPV}(k_p, k_e) = \sum_{i=1}^{T} N \cdot \text{Rev}_i^N(k_p, k_e) \cdot \gamma^i - \nu_p \cdot k_p - \nu_e \cdot k_e, \quad (4)$$

with the input variables shown in the following table Table 3.

Finally, $\text{Rev}_i^N(k_p, k_e)$ denotes the revenue that can be obtained from the storage service for each cycle in year $i$:

$$\text{Rev}_i^N(k_p, k_e) = p^o \cdot \eta \cdot x_i \cdot k_e, \quad (5)$$

with $p^o$ denoting the unit revenue (possibly the price premium) per kWh stored. The discounted value of cash flows obtained from such a battery storage investment can be rewritten as $\Gamma \cdot PM(k_p, k_e)$ where $\Gamma$ denotes a proportionality factor given by $\Gamma \equiv N \cdot \eta \cdot \sum_{i=1}^{T} x_i \cdot \gamma^i$, and PM denotes the profit margin per charging/discharging event:

$$\text{PM}(k_p, k_e) = (p^o - \text{LCOEC}) \cdot k_e - \text{LCOPC} \cdot k_p. \quad (6)$$

Here, LCOEC and LCOPC are the levelized costs of the energy and power components of the storage system, respectively:

$$\text{LCOPC} = \frac{\nu_p}{\Gamma} \qquad \text{LCOEC} = \frac{\nu_e}{\Gamma}.$$

In order for the unit revenue, $p^o$, to be a break-even price, the profit margin per charging/discharging event must be zero, or $p^o = \text{LCOES}\left(\frac{k_e}{k_p}\right)$. Thus the LCOES at the duration $D = \frac{k_e}{k_p}$ is the break-even price per kWh. If there are fixed costs, FC, that are unrelated to the size of the battery, e.g., installation related costs such as permitting and inspection, then:

$$\text{NPV}(k_p, k_e) = \Gamma \cdot \text{PM}(k_p, k_e) - \text{FC}, \quad (7)$$

and the break-even value for a zero net-present value becomes

$$p^o = \text{LCOES}\left(\frac{k_e}{k_p}\right) + \frac{\text{FC}}{k_e \cdot \Gamma}. \quad (8)$$

**Optimal battery size with one representative day.** In connection with a residential solar PV system and a constant unit revenue for electricity stored, there is,

### Table 3 Input variables for the NPV expression

| | |
|---|---|
| $\nu_p$ | System Price of power components (in \$ per kW) |
| $\nu_e$ | System price energy component (in \$ per kWh) |
| $\gamma = \frac{1}{1+r}$ | Discount factor based on the discount rate $r$ (scalar) |
| $T$ | Useful Life of the battery system (in years) |
| $x_i$ | Storage degradation factor (scalar) |
| $\eta$ | Round-trip efficiency factor of the storage system (scalar) |
| $N$ | Number of charge and discharge cycles per year (scalar) |

for any given $k_p$ and $k_e$ no loss of generality in assuming that the system goes through daily charging and discharging events ($N = 365$), even though the battery may not be fully charged. This will generally be the case when there are significant seasonal variations in the household's load and generation profiles. Absent any seasonal variations, an efficient battery system will size the power rating so as to absorb $k_e$ kWh of energy with minimal power. The relation between $k_e$ and $k_p$ is therefore determined by the household's electricity load $L(\cdot)$ and the generation profile of the solar PV system, $G(\cdot)$, as illustrated in Fig. 2. Formally, we again have

$$E^+(k_p) = \int_0^{24} \left[ \min\{L(t) + k_p, G(t)\} - \min\{L(t), G(t)\} \right] dt, \quad (9)$$

and

$$E^-(k_p) = \int_0^{24} \left[ \max\{L(t), G(t)\} - \max\{L(t) - k_p, G(t)\} \right] dt. \quad (10)$$

An optimally sized battery system maximizes the profit margin per cycle PM $(k_p, k_e)$, with $p^o$ given by the levelized price premium:

$$\text{pp} = \frac{1}{\Gamma} \cdot 365 \sum_{i=1}^{T} [x_i \cdot \eta \cdot p - \text{OT}] \cdot \gamma^i, \quad (11)$$

Here, $p$ denotes the retail price of electricity and OT denotes the applicable overage tariff that the customer can obtain for surplus energy fed to the grid. These gains must be adjusted for round-trip energy losses. Thus, an optimal storage system maximizes the daily profit margins (averaged across the years):

$$\text{PM}(k_p) = (\text{pp} - \text{LCOEC}) \cdot E(k_p) - \text{LCOPC} \cdot k_p, \quad (12)$$

where $E(k_p) = \min\{E^+(k_p), E^-(k_p)\}$. Thus $E_s(k_p)$ represents the maximum that can both be charged during the hours of the day with surplus solar power and discharged during the hours when the household's load exceeds the available solar rooftop energy. We note that $E(k_p)$ is concave because the minimum of the two concave functions $E^+(k_p)$ and $E^-(k_p)$ is concave. The derivative of $E(k_p)$ is given by $\mathcal{I}^+(k_p)$ if $E^+(k_p) < E^-(k_p)$, while this derivative is given by $\mathcal{I}^-(k_p^*)$, defined as the length of the time intervals $I^-(k_p) \equiv \{t \in [0, 24] | L(t) - k_p > G(t)\}$, if $E^+(k_p) > E^-(k_p)$. To characterize the power rating of the optimally sized battery system, there are three possible candidates. First if the power capacity $k_p^1$ given by:

$$\text{pp} = \text{LCOES}(\mathcal{I}^+(k_p^1)) \quad (13)$$

is such that $E^+(k_p^1) \le E^-(k_p^1)$, then $k_p^1$ is the optimal power rating. Secondly, if the $k_p^2$ given by:

$$\text{pp} = \text{LCOES}(\mathcal{I}^-(k_p^2)) \quad (14)$$

is such that $E^-(k_p^2) \le E^+(k_p^2)$, then $k_p^2$ is the optimal power rating. If neither of these scenarios apply, there is a unique value $k_p^3$ between $k_p^1$ and $k_p^2$ such that $E^+(k_p^3) = E^-(k_p^3)$. This $k_p^3$ must be the optimal value because, if the first two scenarios do not apply, the function PM $(k_p) = (\text{pp} - \text{LCOEC}) \cdot \min\{E^+(k_p), E^-(k_p)\} - \text{LCOPC} \cdot k_p$ must be increasing up to $k_p^3$ and decreasing thereafter.

**Optimal battery size with seasonal variations.** To incorporate the effect of seasonal variations in both solar PV generation and household consumption, we divide the year into $n$ different seasons, with one representative day for each season. Let $\Delta_s$ represent the share of days out of the total 365 days of the year. Thus the $\Delta_s$ sum up to one. The functions $E^+(k_p)$ and $E^-(k_p)$ in Eqs (9) and (10) are then indexed by $1 \le s \le n$, i.e., $E_s^+(k_p)$ and $E_s^-(k_p)$ Correspondingly, let $E_s(k_p) = \min\{E_s^+(k_p), E_s^-(k_p)\}$. Taking seasonal variations into account, the daily profit margin can be stated as:

$$\text{PM}(k_p, k_e) = \sum_{s=1}^{n} \text{pp} \cdot \Delta_s \cdot \min\{k_e, E_s(k_p)\} - \text{LCOEC} \cdot k_e - \text{LCOPC} \cdot k_p, \quad (15)$$

This objective function reflects that in each season the maximum energy that can be stored is the lower of two values: the energy capacity of the battery and the energy that can effectively be charged and discharged in one day during that particular season, given the power rating, $k_p$. If there is only one season, then $k_e = E(k_p)$. With seasonal variations, the optimal energy capacity will need to balance the value that storage creates in different seasons.

The optimal energy capacity for any given power rating $k_p$ maximizes the objective function in (15) with regard to $k_e$. To that end, we denote by $(m(1), ...., m(n))$ a permutation of the integers $(1, ..., n)$ such that $E_{m(1)}(k_p) \le ..., \le E_{m(n)}(k_p)$. According to the objective function in (15), adding another energy unit to the battery makes sense only to the extent that this additional kWh can be put to use in sufficiently many seasons so that the corresponding revenues collectively outweigh the incremental cost of LCOEC. Formally, for any given $k_p$, the optimal energy capacity $k_e^*(k_p)$ is given by $E_{m(u)}(k_p)$, where $1 \le u \le n$ is determined as the unique

integer satisfying the inequalities:

$$\sum_{s=u}^{n} \mathrm{pp} \cdot \Delta_s \geq \mathrm{LCOEC} \geq \sum_{s=u+1}^{n} \mathrm{pp} \cdot \Delta_s \quad (16)$$

The final step now identifies the optimal power capacity, $k_\mathrm{p}^*$ as the maximizer of:

$$\mathrm{PM}^*(k_\mathrm{p}) = \sum_{s=1}^{n} \mathrm{pp} \cdot \Delta_s \cdot \min\{k_\mathrm{e}^*(k_\mathrm{p}), E_s(k_\mathrm{p})\} - \mathrm{LCOEC} \cdot k_\mathrm{e}^*(k_\mathrm{p}) - \mathrm{LCOPC} \cdot k_\mathrm{p}. \quad (17)$$

**Federal policy support mechanisms**. Germany only provides modest incentives for battery storage installations[33]. Yet, as shown above, our calculations indicate that the large difference between retail rates and feed-in tariffs makes subsidies are no longer necessary for further deployment of behind-the-meter residential storage.

Battery installations in the U.S. qualify for an Investment Tax Credit, ITC, provided the battery can be classified as solar equipment. Formally, the federal tax credit can be represented as:

$$\widehat{\mathrm{ITC}}(k_\mathrm{p}, k_\mathrm{e}|\bar{G}) = .3 \cdot A(k_\mathrm{e}|\bar{G}) \cdot [\nu_\mathrm{p} \cdot k_\mathrm{p} + \nu_\mathrm{e} \cdot k_\mathrm{e}], \quad (18)$$

where $\bar{G}$ denotes the total energy generated on an average day by the solar PV system, that is,

$$\bar{G} = \int_0^{24} G(t)\, dt,$$

and

$$A(k_\mathrm{e}|\bar{G}) = \min\left\{1, \frac{\bar{G}}{k_\mathrm{e}}\right\}$$

provided $\frac{\bar{G}}{k_\mathrm{e}} \geq .75$, while $A(k_\mathrm{e}|\bar{G}) = 0$ otherwise. Thus, the partial tax credit under the ITC drops to zero if more than 25% of the battery's energy storage capacity would be available after absorbing the entire energy generated by the incumbent solar PV system. On a levelized basis, the ITC then modifies the daily profit margin so that:

$$\mathrm{PM}(k_\mathrm{p}, k_\mathrm{e}|\bar{G}) = \sum_{s=1}^{n} \mathrm{pp} \cdot \Delta_s \cdot \min\{k_\mathrm{e}, E_s(k_\mathrm{p})\} - \mathrm{LCOEC} \cdot k_\mathrm{e} - \mathrm{LCOPC} \cdot k_\mathrm{p} \quad (19)$$
$$+ \frac{1}{\Gamma} \cdot \mathrm{ITC}(k_\mathrm{p}, k_\mathrm{e}|\bar{G}).$$

Provided the battery storage system is not oversized relative to the incumbent solar PV system, the ITC will simply scale both LCOPC and LCOEC by the factor .7 (i.e., $.7 = 1-.3$).

**California's SGIP**. California's Self-Generation Incentive Program (SGIP) offers investors a rebate for battery storage systems. The amount to be rebated depends on the duration of the storage system. For a $(k_\mathrm{p}, k_\mathrm{e})$ battery system, the daily profit margin in California, denoted by $\mathrm{PM}^c$, then becomes:

$$\mathrm{PM}^c(k_\mathrm{p}, k_\mathrm{e}|\bar{G}) = \sum_{s=1}^{n} \mathrm{pp} \cdot \Delta_s \cdot \min\{k_\mathrm{e}, E_s(k_\mathrm{p})\} - \mathrm{LCOEC} \cdot k_\mathrm{e} \quad (20)$$
$$- \mathrm{LCOPC} \cdot k_\mathrm{p} + \mathrm{LS}^c = c(k_\mathrm{p}, k_\mathrm{e}).$$

Here, $\mathrm{LS}^c(k_\mathrm{p}, k_\mathrm{e})$ denotes the levelized subsidies available in California for a $(k_\mathrm{p}, k_\mathrm{e})$ battery storage system that qualifies as solar equipment under the ITC rules.

$$\mathrm{LS}^c(k_\mathrm{p}, k_\mathrm{e}) = \frac{1}{\Gamma} \cdot \left[\mathrm{ITC}(k_\mathrm{p}, k_\mathrm{e}|\bar{G}) + \mathrm{SGIP}(k_\mathrm{p}, k_\mathrm{e})\right],$$

with:

$$\mathrm{SGIP}(k_\mathrm{p}, k_\mathrm{e}) = \begin{cases} 400 \cdot k_\mathrm{e} & \text{for } \frac{k_\mathrm{e}}{k_\mathrm{p}} \leq 2 \\ 800 \cdot k_\mathrm{p} + 200(k_\mathrm{e} - 2h \cdot k_\mathrm{p}) & \text{for } 2 \leq \frac{k_\mathrm{e}}{k_\mathrm{p}} \leq 4 \\ 1200 \cdot k_\mathrm{p} + 100(k_\mathrm{e} - 4h \cdot k_\mathrm{p}) & \text{for } 4 \leq \frac{k_\mathrm{e}}{k_\mathrm{p}} \leq 6 \\ 1400 \cdot k_\mathrm{p} & \text{for } \frac{k_\mathrm{e}}{k_\mathrm{p}} \geq 6. \end{cases}$$

## Data availability
The source data underlying Fig. 1, Tables 1 and 2, and Supplementary Figs. 1 and 2 are provided as a Source Data file.

## Code availability
Computational code is available upon request to the corresponding author.

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

## Acknowledgements

The authors gratefully acknowledge financial support from the Sustainable Energy Initiative at the Stanford Graduate School of Business and the Steyer-Taylor Center for Energy Policy and Finance at Stanford University. We are grateful to Gunther Glenk, Felix Baumgarte and three anonymous reviewers for helpful comments and feedback.

## Author contributions

Both authors contributed equally to the development of the research question, approach, and manuscript development. S.R. took the lead in developing the model framework; S.C. took the lead with the empirical analysis.

## Additional information

**Competing interests:** The authors declare no competing interests.

