## [Peer Review File · Nature Communications]

Reviewers' comments:

Reviewer #1 (Remarks to the Author):

Key results: The authors calculate the LCOES of battery systems for the implementation in residential applications in combination with PV. Also, they show a method of optimal sizing of the battery system based on the load curve and the value of electricity.

Abstract:

The abstract should also speak about results and conclusions of the paper.

Introduction

- Headline "Introduction" is missing
- The introduction talks about storage in general, while the paper is about battery storage. When stating the cost this gets specifically important. Also the focus on residential applications should be noted somewhere in the paper.
- 3rd paragraph: Currency is missing, it is only stated cents per kWh, but not which currency.

Results:

- With so very low cost per kWh and so high cost per kW, it is clear that the results show large capacities and low power values. In many other studies only a price per kWh is given, as the difference between batteries with varying power is not so high. This should be double-checked and also discussed in the Discussion part of the paper.
- There are several studies on the optimal PV and battery sizing. The authors need to give a short introduction on them and discuss why the results of the authors are so different from former research.
- P. 9: definition of $CF(t)$ is missing.
- P. 10: Feed-in tariff in Germany is paid for 20 years. Also, I know of no evidence that afterwards the retail electricity price will be paid. This should be marked as an assumption or provided with references.

Diskussion

- This part is very weak. See remarks above.

Methods:

- Methods section needs more references and more information on which methods are available in the literature. The authors should say which approaches their method is based on.

Supplementary notes:

- SN 1.1 it is not clear which one is the “first literature”, “second literature”... I suggest either numbering the sources in the table or naming the authors in the text.
- Is the assumption, that the average decrease of the analyzed years can be extrapolated? Please discuss critically.
- SN 3.3: The input parameters need better sources. Please work with primary sources (e.g. for efficiency). At least for Germany the assumption of 365 cycles per year is too high. It is rather 220.

These sources might be an additional help for you:

Figgenger, Jan; Haberschusz, David; Kairies, Kai-Philipp; Wessels, Oliver; Tepe, Benedikt; Ebbert, Markus et al. (2017): Wissenschaftliches Mess- und Evaluierungsprogramm Solarstromspeicher 2.0. Jahresbericht 2017. Hg. v. RWTH Aachen (0102). <http://www.speichermonitoring.de>.

Weniger, Johannes; Tjaden, Tjarko; Quaschnig, Volker (2014b): Sizing of Residential PV Battery Systems. In: Energy Procedia 46, S. 78–87. DOI: 10.1016/j.egypro.2014.01.160.

Reviewer #2 (Remarks to the Author):

In principle, I like this paper and it touches on a couple very interesting questions around storage cost metrics and optimal sizing. I like that the LCOES metric translates directly to the breakeven price, and that metric seem to make sense. The two use cases picked to look at sizing were also relevant, valuable, and interesting.

I got a bit lost in understanding v_e and v_p and how these were calculated. See comments below about the supplementary section. I think this would need to be clarified before publication, unless I am missing something obvious to the other reviewers. I also had two minor comments about the rest of the text:

-On pg. 6, authors reference other existing storage cost measures, but I think they should mention this and discuss earlier and maybe add a bit more discussion about their approach in the context of other approaches and the literature

-pg 5.

-It's not quite clear to me how that formal statement validates the metric

Notes on Supplementary Materials, ve, vg:

SN: 1.1.

-pg. 1 last sentence, continues on pg 2: this is confusing to me. It sounds like the "first literature" has data from 2017-2025, but a single annual rate of change on this was calculated and used it to do some sort of interpolation to get 2018-2023 data, but I don't understand why that is necessary if the literature spans 2017-2025 already, or why there is just one % annual change used? Based on pg 1, maybe there is a typo and it should say 2013-2023 instead? but even then, ref 4 is a bloomberg source, which I believe contains data from 2010-2016 and then the projections to 2025 and even beyond, so why do they need to interpolate?

-also not clear what "first literature" means - there is a citations but this doesn't match up with the first column in SN1

-Discussion on costs over time are determined could use clarification. It seems weird to linearly interpolate a bunch of sources if there is missing data and then average them (if that is what is being done, it honestly wasn't clear to from the explanation. maybe just needs clarification).

SN: 1.2.

-I believe this section title is mislabeled and it should be "Unit Cost of Power Component". the variables are also mislabeled a couple places and should be doubled checked

-also, what is the unit cost of storage? this is reference in SN1.1. and SN1.2, but is this the same as the unit cost of energy? and if so, are the literature values cited really the unit cost of the energy-related components of the battery only? I didn't have a chance to go through all of them, but at least some of the references look to me like these are total battery costs, not just the kWh-dependent components

-what is the baseline? how was this selected? and why was an average used in ve and a baseline in vp?

-This whole discussion about the “unit costs” (v_e , v_p) is pretty confusing to me and so I can’t really figure out what is being done to tell if it makes sense. It would be really helpful to write out what components are included in each bucket, and provide more clear explanation of how v_e and v_g are determined.

Also, on pg 9 of the SI, it says:

“Certain input parameters contained within Table SN6 require some additional explanation. Regarding the “Deflator” parameter, because of the difference in useful life between the energy and power components, a deflator is applied to the power components such that only a proportion of its total cost is applied to the operational life of the complete system (10 years). “

I don’t understand this. If some components were out at 10 years, they nominally need to be replaced then, and so they should just incur another capital cost associated with that O&M in year 10 of the cashflow. I don’t know what this deflator is or why it is used. There is an equation that is presumably supposed to explain this, but don’t include labels for any of the variables so it’s unclear what this means

Reviewer #3 (Remarks to the Author):

Manuscript NCOMMS-18-24926-T

The emergence of cost effective battery storage

The topic is very interesting and the split of the LCOE into power and energy costs quite interesting. The paper is clear and well written. However, it is less clear when the investment in battery takes place in the household investment decision, probably after installing the PV infrastructure, since the later does not interfere in the optimization program. Since the same operator (the household) holds both (PV + battery), why the size of the panel is considered as given? And how accounting for both PV + battery could change the investment program?

The size of the battery is highly dependent here on the storage duration. Clarification should be made on the optimization program which is built on daily basis (DPM, eq. 2). Usually the interval is

longer to account for the variations in both solar energy potential and the load curve. Extending this interval to several days would change results? And how?

A strong assumption in this model is that the surplus is sold to the grid (area III in Fig 2), while in practice for massive PV deployment and cumulative excess on the grid due to self-consumption, the demand from the grid should be low down to zero, especially in areas with congestion at mid-day, as in Germany. Making a business model based on the volumes sold to the grid at high tariffs seems risky. Sizing the battery could be done in a first time for self-consumption, and secondly selling the excess when possible, because the regulation is changing quite often and tariffs as well.

The use of NPV is quite surprising (page 15) since the household objective is not to become a service provider, or at least here it was not mentioned at the beginning. If battery is installed to this effect then sizing is indeed affected; but then ancillary services and market segments would need to be presented in detail.

While methodology seems robust with respect to LCOE, it is less convincing while accounting for the NPV, as self-consumption is not valued and profits are based on revenues on the power market, which are somehow overestimated with respect to volumes and tariffs. The study case is also less convincing, due to congestions issues that DSOs face in some areas in Germany. By specifying where this model could apply, it could help understanding the type of households who could adopt this rational.

Minor comments

In the paper structure, assumptions needs to be isolated from the section of results. For instance the section "The Cost of energy" is a mix of both hypothesis and results and it is difficult to understand where the authors' calculus are and where the data is an input.

My black and white print could not follow explanations of curves by color, make sure that when you publish your work this issue is solved.

Page 4-5: LCOE units of measure should necessary be expressed in monetary units and not in kWh or kW.

Wording: page 4 "lifetime of the battery time", page 5 "power-and", page 11 "service service", page 20 "the becomes"

Reviewer #1 (Remarks to the Author):

Key results: The authors calculate the LCOES of battery systems for the implementation in residential applications in combination with PV. Also, they show a method of optimal sizing of the battery system based on the load curve and the value of electricity.

Abstract:

The abstract should also speak about results and conclusions of the paper.

Thank you, we agree. You will find that the revised Abstract seeks to highlight our results and conclusions, though we needed to stay within the journal's word limit on Abstracts.

Introduction

- *Headline "Introduction" is missing*

The journal's style guidelines instruct authors to proceed without a headline "Introduction".

• *The introduction talks about storage in general, while the paper is about battery storage. When stating the cost this gets specifically important. Also the focus on residential applications should be noted somewhere in the paper.*

The revised version has been more careful in emphasizing that the initial part of LCOES is generically about the cost of storage. We then present some numerical results about the cost of battery storage and then proceed to optimized lithium-ion battery storage in conjunction with solar PV in the final part.

- *3rd paragraph: Currency is missing, it is only stated cents per kWh, but not which currency.*

Thank you. We have sought to indicate the applicable currency at all stages of the paper.

Results:

• *With so very low cost per kWh and so high cost per kW, it is clear that the results show large capacities and low power values. In many other studies only a price per kWh is given, as the*

difference between batteries with varying power is not so high. This should be double-checked and also discussed in the Discussion part of the paper.

Indeed, we believe that any cost calculation for storage based only on kWh, and without reference to the power capacity of the system, is incomplete and or misleading in terms of the economics of storage. The revised version makes this point more explicit both in the Results and in the Discussion sections.

- *There are several studies on the optimal PV and battery sizing. The authors need to give a short introduction on them and discuss why the results of the authors are so different from former research.*

In the Results section, we have added additional references (on page 4) –including the ones you kindly provided- and expanded our discussion of how these earlier studies on storage relate to our approach (page 5, second paragraph, and pages 6 -7) and specifically to optimal PV and battery sizing in the final paragraph (page 10).

- *P. 9: definition of $CF(t)$ is missing.*

Yes, this was a typo. Instead of $CF(t)$, it should have said $G(t)$ which had been introduced in connection with Figure 2.

- *P. 10: Feed-in tariff in Germany is paid for 20 years. Also, I know of no evidence that afterwards the retail electricity price will be paid. This should be marked as an assumption or provided with references.*

Thank you for pointing out that the feed-in-tariff is available for 20 rather than 15 years in Germany. This correction does actually not affect our results since our calculations assume that the solar PV system is eligible for the feed-in-tariff throughout the 10-year useful life of the lithium-ion battery system. For the same reason, it does not matter for our results as to what revenue the household receives for surplus energy after the initial feed-in-tariff period. In response to the reviewer's comment we have changed and clarified the wording in the text on page 11, penultimate paragraph.

Diskussion

- *This part is very weak. See remarks above.*

The Discussion section has been revised and expanded significantly.

Methods:

- *Methods section needs more references and more information on which methods are available in the literature. The authors should say which approaches their method is based on.*

We have done so. In particular, we have elaborated (on page 18) that our method is the standard microeconomic approach positing that the household will seek to install a battery system that maximizes the discounted value of cash flows associated with the investment.

Supplementary notes:

- *SN 1.1 it is not clear which one is the “first literature”, “second literature”... I suggest either numbering the sources in the table or naming the authors in the text.*

Thank you. We have comprehensively revised the Supplementary Note, including appropriately labeling the different literature sources throughout.

- *Is the assumption, that the average decrease of the analyzed years can be extrapolated? Please discuss critically.*

We seek to estimate a trajectory for the unit costs of power and energy storage components, highlighting the historical cost reductions from 2013 to 2017, and aggregating projections from the literature to project the years from 2018 to 2023.

For the energy components, v_e , the historical costs are based on literature sources (2) – (7) as labeled in the revised SN (pages 2-3). To be clear, only the 2017 values or earlier from each of these literature sources are used for historical costs. The projected costs – 2018 to 2023 – are based on literature sources (4) – (8). To determine the individual year projections for each source provided (again, (4) – (8)) for 2018 to 2023, the detailed procedure is described in the Supplementary Note, pages 1-3.

For the unit power components, the procedure used to determine historical and projected costs is given on pages 6-7 of the Supplementary Note.

The overall annual decrease in the LCOES makes use of the values provided in Tables SN2 and SN3 – unit energy and power components respectively – for each duration given in Figure 1 of the revised main text. The data that sits behind Figure 1 of the main text is given in the “Source Data” Excel file. Finally we provide a brief comment on the relative conservative estimate of projected unit energy component cost reductions, especially given the anticipated expansion of battery manufacturing capacity, on pages 4-5 of the Supplementary Note.

- *SN 3.3: The input parameters need better sources. Please work with primary sources (e.g. for efficiency). At least for Germany the assumption of 365 cycles per year is too high. It is rather 220.*

We have included additional references to Tables SN6 (Germany) and SN9 (California). These references contain estimates of the relevant parameters in the academic literature. With respect to number of cycles per year, earlier studies appear to have viewed this as a modeling choice; see references in Tables SN6 and SN9 in the Supplementary Note. In the specific case of Germany, given the average existing solar facility installation (6 kWp), load pattern and flat tariff structure that provides no differentiation between weekdays and weekend days, a 365 cycling assumption appears appropriate. It is important to note that cycling in this case does not imply full energy capacity charge and discharge per day, rather that only some proportion of full charge/discharge cycle will occur. Depending on the season, it may be the case that the battery does not fully discharge the energy charged. We have clarified this point on page 5, second paragraph, and on page 19, penultimate paragraph in the main text.

These sources might be an additional help for you:

Figgenger, Jan; Haberschusz, David; Kairies, Kai-Philipp; Wessels, Oliver; Tepe, Benedikt; Ebbert, Markus et al. (2017): Wissenschaftliches Mess- und Evaluierungsprogramm Solarstromspeicher 2.0. Jahresbericht 2017. Hg. v. RWTH Aachen (0102). <http://www.speichermonitoring.de>.

Weniger, Johannes; Tjaden, Tjarko; Quaschnig, Volker (2014b): Sizing of Residential PV Battery Systems. In: Energy Procedia 46, S. 78–87. DOI: 10.1016/j.egypro.2014.01.160.

Thank you, we have incorporated these references in the revised manuscript and the Supplementary Note.

Reviewer #2 (Remarks to the Author):

In principle, I like this paper and it touches on a couple very interesting questions around storage cost metrics and optimal sizing. I like that the LCOES metric translates directly to the breakeven price, and that metric seem to make sense. The two use cases picked to look at sizing were also relevant, valuable, and interesting.

I got a bit lost in understanding v_e and v_p and how these were calculated. See comments below about the supplementary section. I think this would need to be clarified before publication, unless I am missing something obvious to the other reviewers. I also had two minor comments about the rest of the text:

-On pg. 6, authors reference other existing storage cost measures, but I think they should mention this and discuss earlier and maybe add a bit more discussion about their approach in the context of other approaches and the literature

The revised version does so briefly on page 5 so as to maintain the flow of the argument we are making. We have expanded our discussion of existing alternative cost measures on page 6-7.

-pg 5.

-It's not quite clear to me how that formal statement validates the metric

Our wording here was misleading and we corrected it. The formal statement makes a specific claim that is then validated in the Methods section.

—
Notes on Supplementary Materials, v_e , v_g :

SN: 1.1.

-pg. 1 last sentence, continues on pg 2: this is confusing to me. It sounds like the "first literature" has data from 2017-2025, but a single annual rate of change on this was calculated and used it to do some sort of interpolation to get 2018-2023 data, but I don't understand why that is necessary if the literature spans 2017-2025 already, or why there is just one % annual change used? Based on pg 1, maybe there is a typo and it should say 2013-2023 instead? but even then, ref 4 is a bloomberg source, which I believe contains data from 2010-2016 and then the projections to 2025 and even beyond, so why do they need to interpolate?

We have rewritten the Supplementary Note substantially to improve clarity. The need for interpolation for some of the literature sources used is to create annual energy component cost estimates for each of literature sources used. This then allows us to make corresponding cost projections. This procedure is explained in greater depth, specifically on pages 1-3 of the Supplementary Note.

-also not clear what "first literature" means - there is a citations but this doesn't match up with the first column in SN1

Yes, we have now labeled the different literature sources consistently throughout the Supplementary Note.

-Discussion on costs over time are determined could use clarification. It seems weird to linearly interpolate a bunch of sources if there is missing data and then average them (if that is what is being done, it honestly wasn't clear to from the explanation. maybe just needs clarification).

We agree that the explanation should be clearer and we have rewritten the relevant cost projection procedure on pages 1-3 of the Supplementary Note.

SN: 1.2.

-I believe this section title is mislabeled and it should be "Unit Cost of Power Component". the variables are also mislabeled a couple places and should be doubled checked

Thank you, we have corrected several typos.

-also, what is the unit cost of storage? this is reference in SN1.1. and SN1.2, but is this the same as the unit cost of energy? and if so, are the literature values cited really the unit cost of the energy-related components of the battery only? I didn't have a chance to go through all of them, but at least some of the references look to me like these are total battery costs, not just the kWh-dependent components

We have clarified the language used in the Supplementary Note. To be clear, all literature sources labeled (1) – (8) on pages 1-3 of the Supplementary Note pertain to the energy components of the battery storage system. The definition of energy and power components is provided in the opening paragraph to the Supplementary Note on page 1.

-what is the baseline? how was this selected? and why was an average used in v_e and a baseline in v_p ?

We have provided a more comprehensive and detailed explanation of how the historical and projected unit cost of power components, v_p , has been calculated on pages 6 – 7 of the revised Supplementary Note.

-This whole discussion about the "unit costs" (v_e , v_p) is pretty confusing to me and so I can't really figure out what is being done to tell if it makes sense. It would be really helpful to write out what

components are included in each bucket, and provide more clear explanation of how v_e and v_g are determined.

We have taken care in rewriting pages 1 – 7 of the Supplementary Note to increase the clarity of the terms and the approach we have taken.

Also, on pg 9 of the SI, it says:

“Certain input parameters contained within Table SN6 require some additional explanation. Regarding the “Deflator” parameter, because of the difference in useful life between the energy and power components, a deflator is applied to the power components such that only a proportion of its total cost is applied to the operational life of the complete system (10 years). “

I don’t understand this. If some components were out at 10 years, they nominally need to be replaced then, and so they should just incur another capital cost associated with that O&M in year 10 of the cashflow. I don’t know what this deflator is or why it is used. There is an equation that is presumably supposed to explain this, but don’t include labels for any of the variables so it’s unclear what this means.

The approach the Reviewer sketches would lead to results that should be principally equivalent for an analysis if one were to consider a 30-year horizon. The energy storage modules would then be replaced twice since they have useful life of only 10 years. Our calculations focus on a horizon of only 10 years and assume that the power component, which then still has a remaining useful life of 20 years, can be redeployed. The deflator determines the percentage of the system price of the power component that is to be charged for a capital expenditure that commits the asset for only 10 years. The revised Supplementary Note on pages 11 – 12 seeks to make this approach clear.

Reviewer #3 (Remarks to the Author):

Manuscript NCOMMS-18-24926-T

The emergence of cost effective battery storage

The topic is very interesting and the split of the LCOE into power and energy costs quite interesting. The paper is clear and well written. However, it is less clear when the investment in battery takes place in the household investment decision, probably after installing the PV infrastructure, since the later does not interfere in the optimization program. Since the same operator (the household) holds both (PV + battery), why the size of the panel is considered as given? And how accounting for both PV + battery could change the investment program?

As we now explain in several places of the revised manuscript, our analysis is confined to the addition of a battery storage system that complements an existing solar PV system. We argue that that this is a timely and relevant question for many solar PV customers in many countries. The question you raise we regard as a natural next step, and say so in the Discussion section. To be sure, the joint optimization of a solar PV + Battery system will involve a significantly broader analytical framework. Based on some preliminary work we have done, this is doable but will take us significantly beyond the scope of the current analysis. Our objective in the current paper was to introduce the levelized cost metric and show its usefulness in determining optimally sized battery systems.

The size of the battery is highly dependent here on the storage duration. Clarification should be made on the optimization program which is built on daily basis (DPM, eq. 2). Usually the interval is longer to account for the variations in both solar energy potential and the load curve. Extending this interval to several days would change results? And how?

As we argue in the main text on page 19, for any given battery configuration that is used in conjunction with a solar PV system, there is in the context of our analysis no loss of generality in assuming daily charge and discharge cycles, though the battery may not be discharged fully on any given day. Our model accounts for variations in the solar energy generation by distinguishing between summer and winter days. Additional random variations in generation and load curve could be included in the model, but at this point we do not know whether this would have generalizable predictive effect on the optimal size of the battery.

A strong assumption in this model is that the surplus is sold to the grid (area III in Fig 2), while in practice for massive PV deployment and cumulative excess on the grid due to self-consumption, the demand from the grid should be low down to zero, especially in areas with congestion at mid-day, as in Germany. Making a business model based on the volumes sold to the grid at high tariffs seems risky. Sizing the battery could be done in a first time for self-consumption, and secondly selling the excess when possible, because the regulation is changing quite often and tariffs as well. The use of NPV is quite surprising (page 15) since the household objective is not to become a service provider, or at least here it was not mentioned at the beginning. If battery is installed to this effect then sizing is indeed affected; but then ancillary services and market segments would need to be presented in detail.

The revised paper argues in more detail that the optimizing household seeks to maximize the future discounted cash flows (NPV) associated with the battery investment decision. The Reviewer is absolutely correct that our representative household does not act as a service provider and the only source of revenue is the avoided cost of the retail rate. In our applications for Germany and California, the household also knows the rate at which surplus power from the solar system will be credited.

While methodology seems robust with respect to LCOE, it is less convincing while accounting for the NPV, as self-consumption is not valued and profits are based on revenues on the power market, which are somehow overestimated with respect to volumes and tariffs. The study case is also less convincing, due to congestions issues that DSOs face in some areas in Germany. By specifying where this model could apply, it could help understanding the type of households who could adopt this rational.

We agree that issues of congestion and the value of self-consumption are not considered as part of the current analysis. We focus on an individual household that makes the battery storage decision based on its own tradeoff between upfront investment expenditure and ongoing revenues determined by the retail rate and the overage tariff (feed-in-tariff in Germany; retail rate minus 2.7 U.S. cents per kWh in California).

Minor comments

In the paper structure, assumptions needs to be isolated from the section of results. For instance

the section “The Cost of energy” is a mix of both hypothesis and results and it is difficult to understand where the authors’ calculus are and where the data is an input.

Thank you for this general point. We have tried to be clearer throughout the *Results* section as to what is a modeling assumption, what are generic findings of the model analysis and what are numerical calibrations based on literature sources (Supplementary Notes).

My black and white print could not follow explanations of curves by color; make sure that when you publish your work this issue is solved.

We have changed the labeling of the curves so they can be interpreted both with and without color.

Page 4-5: LCOE units of measure should necessary be expressed in monetary units and not in kWh or kW.

We hope to have addressed all issues associated with monetary units.

Wording: page 4 “lifetime of the battery time”, page 5 “power-and”, page 11 “service service”, page 20 “the becomes”

Thank you, we have addressed these typos.

Reviewers' comments:

Reviewer #1 (Remarks to the Author):

The authors have significantly improved the paper in this revision. However, two major issues remain:

Supplementary notes 1 on page 1 state that the power components refer to the BOS, including all BOS components as well as construction and installation. This definition is very questionable, as the cost of installation will not increase much with a higher power, especially in the household sector where the choice of power is limited. The authors state this approach is used in source 24. I don't see the evidence for that in the named source. It is not clear what is the "technology scope of electrical energy storage technologies".

In SN 1.4 the authors state that the Lazard 4.0 study "views several battery system components as energy rather than power components". The authors say their approach is to use them as fixed costs, but as far as I understand the text they use them as cost per kW. Very high cost per kW are not realistic. This is why the optimized system in Germany has a very low power (below 2) and a very high energy content of more than 12, still resulting in 8.7 ct/kWh LCOES. This is very contradictive to what we see on the market. It is questionable how this large storage capacity is to be filled with this very low power, needing about 9 hours of 1.35 kW PV electricity excess to fully fill the storage which should be the case only on very few days each year. How can 365 cycles be reached then? As far as I understand the full load cycles are used in the calculation for the LCOES, this is why the value is so low.

The definition of the yearly cycles is not clear. In the revision the authors say that the cycles are not full load cycles, but rather the number of days that the system is used. However, in the NPV (Revenues) calculation the number of cycles is multiplied with the system power if I understand it right. This would then assume full load cycles. With the calculated system of 12 kWh and 1.35 kW however, 365 full load cycles are not realistic.

Several equations are not numbered.

Reviewer #2 (Remarks to the Author):

Thank you for incorporating the comments. The revised draft addressed my concerns. - Kelsey Horowitz

Reviewer #3 (Remarks to the Author):

Manuscript NCOMMS-18-24926-T

The emergence of cost effective battery storage

Comments have not been entirely raised in this new version, although the paper seems to better justify the method and the assumptions. It still remains unclear on the side of the consumer's program, whether the consumer becomes independent from the grid, without any withdrawal, and how the battery is charged/ discharged. Fig 2 remains unclear, with respect to area I in particular: the text presents this area as being PV self-consumption ("Absent any battery storage..denoted by p"); while the Fig 2 presents the area I as being the battery self-consumption. The program becomes confusing while contemplating the solar profile which here appears smooth while in practice it is highly intermittent. Fig 2 shows the PV generation starting with 3 am until 8 pm, which is rather optimistic. Data could be referenced and explained if smooth production is due to the aggregation among all households in Munich for instance, or if there is an average, otherwise

results are overestimated.

The intermittency will affect the fatigue of the battery, here the number of cycles and any start ups/ shut downs are not discussed. The "duration" is used instead, and remains unclear if it is charge or discharge. Charging is highly intermittent due to (PV net of self-consumption), the same for the discharge. All curves are here smooth, thus the database is not realistic. All days in Munich seem sunny and this energy potential is obviously overestimated. The conclusion on the economic viability in Germany is highly contestable. If other studies have similar results, the authors could include them. The review at page 10 does not provide any detail if the paper is in line with those studies.

As already commented for the first version, the grid DSO/TSO will decrease the demand at mid-day if "many commercial and residential investors in the future" will invest in such equipment as stated in conclusion. Extrapolating overestimated results to an entire area seems unrealistic. Some sensitivities would be necessary.

Minor comments

The abstract could mention if self-consumption is ensured, and could also give orders of magnitude.

Wording needs to be checked (page 6 "be indeed be", page 7 "have frequently have", page 15 "one one hour" etc).

Reviewer #1 (Remarks to the Author):

The authors have significantly improved the paper in this revision. However, two major issues remain:

Supplementary notes 1 on page 1 state that the power components refer to the BOS, including all BOS components as well as construction and installation. This definition is very questionable, as the cost of installation will not increase much with a higher power, especially in the household sector where the choice of power is limited. The authors state this approach is used in source 24. I don't see the evidence for that in the named source. It is not clear what is the "technology scope of electrical energy storage technologies".

The introductory paragraph of the Supplementary Notes has been expanded to state clearly that we follow the definition of technology scope of electrical storage technologies, as stated in Schmidt et al. (2017).

In SN 1.4 the authors state that the Lazard 4.0 study "views several battery system components as energy rather than power components". The authors say their approach is to use them as fixed costs, but as far as I understand the text they use them as cost per kW. Very high cost per kW are not realistic. This is why the optimized system in Germany has a very low power (below 2) and a very high energy content of more than 12, still resulting in 8.7 ct/kWh LCOES. This is very contradictive to what we see on the market. It is questionable how this large storage capacity is to be filled with this very low power, needing about 9 hours of 1 .35 kW PV electricity excess to fully fill the storage which should be the case only on very few days each year. How can 365 cycles be reached then? As far as I understand the full load cycles are used in the calculation for the LCOES, this is why the value is so low.

Our previous language regarding fixed costs was misleading. We have clarified and the language and substantively also expanded the cost model for lithium ion batteries to include the possibility

of fixed costs that do not vary with either the power rating or the energy storage capacity of the system. Certain costs related to the installation of the battery, e.g., permitting and inspection would be examples of such fixed costs. Please refer to page 6 and 19 in the revised manuscript.

Regarding our findings for a representative household in Munich, we now report the following key results: 0.73 kW, 5.1 kWh a duration of $D=7$ hours and an LCOES of 0.085 € per kWh. To be sure, the LCOES figure does not include any fixed costs, however fixed costs are included in the NPV calculations— see equation 7 on page 19 of the revised manuscript. The new numbers reflect that we now seek to optimize the battery on the basis of representative days for each of the twelve months of the year. Also, our model and calculations now explicitly recognize the applicable constraints in terms of charging and discharging the battery during different seasons. Those constraints in conjunction with monthly production and consumption data lead us to predict a smaller battery system. However, the duration (the ratio of the energy capacity to the power rating) relative to our previous calculation does not change much.

The reviewer raises the point that the battery size we predicted previously is not consistent with observed market practice; we believe the revised results are more consistent with current deployments. Further, as we note in the paper, households may choose among the battery systems available on the market for reasons that go beyond the price arbitrage motive we focus on. Importantly, we also note that the household could effectively double the size of the battery relative to what we predict as optimal, and still be “in the money”, i.e. $NPV > 0$.

The definition of the yearly cycles is not clear. In the revision the authors say that the cycles are not full load cycles, but rather the number of days that the system is used. However, in the NPV (Revenues) calculation the number of cycles is multiplied with the system power if I understand it right. This would then assume full load cycles. With the calculated system of 12 kWh and 1.35 kW however, 365 full load cycles are not realistic.

The revised version seeks to be clear that the battery goes through 365 cycles per year, but some of these will not be full load cycles, that is, the battery is not fully charged, either because there is not enough solar power generated or the surplus energy generated could not possibly be discharged in the hours after sunset. Please see page 10 in the revised manuscript. The revenues derived from the battery system account for the fact that during certain months there will only be partial load cycles. We have created Table 1 for Germany and Table 2 for California in order to track the pattern of full load cycles for the twelve months of the year (see table row “Charge to Capacity?” in both Table 1 and 2 in the revised manuscript).

Several equations are not numbered.

We have numbered only those equations that we later on refer to. It is our understanding that this is consistent with the style guidelines of the journal.

Reviewer #2 (Remarks to the Author):

Thank you for incorporating the comments. The revised draft addressed my concerns. - Kelsey Horowitz

Reviewer #3 (Remarks to the Author):

Comments have not been entirely raised in this new version, although the paper seems to better

justify the method and the assumptions. It still remains unclear on the side of the consumer's program, whether the consumer becomes independent from the grid, without any withdrawal, and how the battery is charged/ discharged. Fig 2 remains unclear, with respect to area I in particular: the text presents this area as being PV self-consumption ("Absent any battery storage..denoted by p"); while the Fig 2 presents the area I as being the battery self-consumption. The program becomes confusing while contemplating the solar profile which here appears smooth while in practice it is highly intermittent. Fig 2 shows the PV generation starting with 3 am until 8 pm, which is rather optimistic. Data could be referenced and explained if smooth production is due to the aggregation among all households in Munich for instance, or if there is an average, otherwise results are overestimated.

The revised version seeks to be clear during which months the household is "grid-positive", that is, it exports power to the grid, but at no point in time buys power from the grid. Figure 2 and the surrounding explanations have been modified. Our model presumes that investors will base their value calculations on average insolation values for a representative day of a particular month.

Regarding the illustrative representation of solar power generation in Figure 2, the first instance of any generation is at 05:00 and the last instance occurs at 19:00. The instantaneous generation at these times is exceedingly low, especially at 05:00 a.m., during times of civil twilight – when the sun is less than 6 degrees below the horizon. From May-July in Munich, civil twilight begins before 5:00 a.m.

The intermittency will affect the fatigue of the battery, here the number of cycles and any start ups/ shut downs are not discussed. The "duration" is used instead, and remains unclear if it is charge or discharge. Charging is highly intermittent due to (PV net of self-consumption), the same for the discharge. All curves are here smooth, thus the database is not realistic. All days in Munich seem sunny and this energy potential is obviously overestimated. The conclusion on the economic viability in Germany is highly contestable. If other studies have similar results, the authors could include them. The review at page 10 does not provide any detail if the paper is in line with those studies.

We hope the revised version is more explicit that the battery will start charging once the generation curve exceeds the load curve (points in time beyond t^* in Figure 2). We expect there to be variations in the amount of solar power generated in real time, but we also expect the battery to remain in "charge mode" until such time as either the energy capacity of the battery has been reached or there is no further need to charge because of a limit in the amount of energy that can be discharged in the hours after sunset. The *Methods* section revised version is now far more explicit on the resulting constraints.

By explicitly including data from 12 different months of the year, we incorporate monthly variations and our calculations reflect that not "all days in Munich are sunny". In fact, relative to the parameters that apply to individual months of the year, our optimal battery is clearly oversized in some months, while it is undersized in others. This is now made explicit in the Table 1.

As already commented for the first version, the grid DSO/TSO will decrease the demand at mid-day if "many commercial and residential investors in the future" will invest in such equipment as stated in conclusion. Extrapolating overestimated results to an entire area seems unrealistic. Some sensitivities would be necessary.

We have modified our wording on that particular point in the Conclusion. At the same time, we note in passing that if the overage tariff were to be reduced at midday, it would only increase the price premium and therefore the value of behind-the-meter battery storage.

Minor comments

The abstract could mention if self-consumption is ensured, and could also give orders of magnitude.

Wording needs to be checked (page 6 “be indeed be”, page 7 “have frequently have”, page 15 “one one hour” etc).

We addressed these minor issues.

REVIEWERS' COMMENTS:

Reviewer #3 (Remarks to the Author):

The revised version is clearer. Minor clarifications could be done at Fig 2 where area V is presented as purchase from the grid in the notes of the title (p. 9) but it is also presented as derived from the area II (do you mean through the grid and back to the household? Please clarify). The area II is necessarily larger than the area IV due to round trip efficiency losses, it could be inserted in the text where authors mention that "region II and IV (insert the plural for "region" please!) have the same area", which is not correct.

Table 1: unclear "Charge to capacity?" what that is mean? Authors could insert a note for table reading.

Typing: check "coupled" p. 12, "couple" maybe?

Page 16: the use of both dollar and euro currencies is confusing, especially when expressed in cents only; it is not the same market or what that is mean?

I couldn't check the references, the version I received has ??? instead of bibliography figures.

Reviewer #3 (Remarks to the Author):

The revised version is clearer. Minor clarifications could be done at Fig 2 where area V is presented as purchase from the grid in the notes of the title (p. 9) but it is also presented as derived from the area II (do you mean through the grid and back to the household? Please clarify).

We have added clarifications on page 7 of the revised manuscript, specifically the text: Region V is residual demand that would not be met by the battery and must be met through purchases from the grid at the going retail rate p .

The area II is necessarily larger than the area IV due to round trip efficiency losses, it could be inserted in the text where authors mention that “region II and IV (insert the plural for “region” please!) have the same area”, which is not correct.

We have updated our language on page 7 of the revised manuscript, specifically the text: region II is surplus energy that can be stored and subsequently discharged as region IV (minus efficiency losses)

Table 1: unclear “Charge to capacity?” what that is mean? Authors could insert a note for table reading.

We have updated the captions for both Tables 1 and 2 to re-emphasize the definition of “Charge to capacity”

Typing: check “coupled” p. 12, “couple” maybe?

We have changed the wording to make the exposition clearer.

Page 16: the use of both dollar and euro currencies is confusing, especially when expressed in cents only; it is not the same market or what that is mean?

There are two markets that are being discussed in the paragraph in question, thus the need for two currencies. We have adjusted the language to make this distinction clearer, while maintaining the two currencies in the text of the revised manuscript.

I couldn't check the references, the version I received has ??? instead of bibliography figures.

We were unable to replicate the error as all references compiled properly. Moreover, upon checking the submission documents, there did not seem to be references that were undefined. We have taken care in this revision to ensure no references are broken.